# Frequency and characteristics of asthma in smokers attending smoking cessation units in Spain

**Juan-Antonio Riesco** [1]* , **Carlos Rábade** [2], **Jaime Signes-Costa** [3], **Eva Cabrera** [4], **Carlos-A Jimenez** [5]

**1** Pulmonology Department, Hospital Universitario San Pedro de Alcántara, Cáceres, Spain, **2** Pulmonology Department, Hospital Universitario Santiago de Compostela, Santiago de Compostela, Spain, **3** Pulmonology Department, Hospital Clínico Universitario de Valencia, Valencia, Spain, **4** Pulmonology Department, Hospital Universitario Virgen de la Victoria, Málaga, Spain, **5** Smoking Cessation Clinic, Hospital Clínico San Carlos, Madrid, Spain

☯ These authors contributed equally to this work.
* jantonio.riesco@gmail.com

**Data Availability Statement:** We have updated an anonymized data file for public sharing in the OSF platform: https://osf.io/zaw92/.

## Abstract

### Introduction

The interaction between smoking and asthma impairs lung function and increases airflow obstruction severity. The identification of smoking patterns in smokers with and without asthma is crucial to provide the best care strategies. The aims of this study are to estimate asthma frequency, describe asthma features, and characterize smoking in smokers attending smoking cessation units.

### Material and methods

We carried out a cross-sectional study in five smoking cessation units with different geographical distribution to estimate asthma frequency in smokers, characterize asthma features in smokers, as well as smoking in asthmatic smokers.

### Results

Asthma frequency among smokers was 18.6%. Asthmatic smokers presented high passive exposure, low smoking self-efficacy and will to quit smoking, as well as a high exacerbation frequency, severe symptoms, and frequent use of long-acting beta agonists, inhaled steroids, and short-acting beta agonists.

### Discussion

Smokers with asthma constitute a high-risk group with worsened evolution of pulmonary involvement. All smokers should be regularly screened for asthma. Effective smoking cessation strategies should be proposed to smokers with asthma in order to reverse the harmful effects of smoking on the airway, together with a comprehensive and integral approach.

**Funding:** This study was supported by a grant of Menarini. The funders had no role in study design, data collection and analysis, decision to publish, or preparation of the manuscript.

**Competing interests:** The authors have declared that no competing interests exist.

## Introduction

Tobacco smoking is the leading cause of preventable premature mortality in the world [1]. Indeed, tobacco smoke contains multiple toxic substances with inflammatory, cytotoxic, mutagenic, and carcinogenic properties. These substances produce direct cellular injuries and chemical reactions, with release of oxygen radicals. Oxidative stress activates epithelial cells which produce inflammatory mediators, and in turn activate neutrophils, macrophages, and cytotoxic T cells. In addition, cigarette smoke impairs innate immune responses mediated by epithelial cells, alveolar macrophages, and dendritic cells [2, 3].

On the other hand, asthma is a chronic airway disease, affecting around 7.5% of the adult population, characterized by variable airway obstruction, airway hyperresponsiveness, and airway inflammation [4]. It is important to mention that asthmatic patients start and continue smoking with prevalence rates relatively close to those found in the general population [5]. Smoking prevalence in asthmatic patients is as high as 25% in developed countries [5–7] and one in four asthmatic patients continues to smoke despite their medical condition [8, 9].

In this context, the physiopathological and clinical interactions between smoking and asthma have been analyzed in different studies, suggesting an "asthma smoking phenotype" [2, 3, 10–14]. Smoking asthmatics have different kinds of bronchial inflammation compared with non-smoking asthmatics [14]. Cigarette smoke induces important changes in the asthma endotype, with a predominance of activated macrophages and neutrophils in the sputum, airways, and lung parenchyma as in early chronic obstructive pulmonary disease [3, 12, 13]. The persistent exposure to cigarette smoke has been shown to drive additive or synergistic inflammatory and remodeling responses (*e.g.*, chronic mucus hypersecretion) in the asthmatic airways, explaining the reported accelerated lung function decline and increased airflow obstruction severity [3, 12, 13].

Cigarette smoking and asthma negatively impact patient's clinical, prognostic, and therapeutic outcomes, regardless of disease severity [8]. Morbimortality in asthma has been depicted as higher in smokers [6, 10]. Asthmatic smokers present more severe symptoms, a worse quality of life, and a higher number of medical visits and hospital admissions compared with non-smokers or ex-smokers [10, 15]. Corticosteroids' response is also impaired among smokers [16]. Smoking cessation improves lung function in patients with asthma, demonstrating that the detrimental effect of smoking on the airway is reversible in asthmatic patients [17].

However, all of the results exposed above have not always been consistent, mainly due to substantial methodological differences [18]. Therefore, it is vital to identify differential smoking patterns between asthmatic and non-asthmatic smokers in order to provide the best care strategies.

We designed this study with the following objectives: 1) estimate asthma frequency in smokers attending smoking cessation units, 2) describe asthma features in smokers, and 3) characterize smoking in asthmatic *vs*. non-asthmatic smokers.

## Material and methods

### Study design and setting

A cross-sectional study was carried out in accordance with the basic ethical principles stated in the Declaration of Helsinki of the World Medical Association on Ethical Principles for Medical Research Involving Human Subjects and its subsequent revisions.

### Ethics statement

The present study was approved by the Committee on the Ethics of Research involving Medicinal Products of all participating centers, according to the Spanish law, before the beginning of the study. All participants signed an informed consent form.

## Participants

We selected smokers attending smoking cessation units in Spain. Included patients were adult smokers, regardless of whether they had an asthma diagnosis or not, and of whether it was their first visit to the smoking unit or not.

A total of five smoking cessation units certified by the Spanish Society of Respiratory Pathology (SEPAR), with different geographical distribution, participated in the study.

A systematic sampling was performed in each smoking cessation unit. Sample size estimation was based on a prevalence of asthma in smokers of 30%, with a confidence level of 95% and a precision of 5%, resulting in a sample size of 340 subjects (68 in each cessation unit). The date range in which the participant recruitment was carried out was November 2021 –July 2022.

## Measurements

The main outcome was the presence of asthma, meaning that there was a documented asthma diagnosis in the medical history of the smoker from the pneumology or allergology departments.

## Descriptive variables

The following variables were collected in all participants: sociodemographic characteristics such as age, gender, educational level (no education/primary, high school, elementary professional training, superior professional training, university degree), employment status (unemployed, active, retired, student, housewife), residence (urban or residential area), physical activity (none, mild defined as $<5$ times/week/at least 30 minutes, moderate as $\geq 5$ times/week/at least 30 minutes, and intense as $\geq 3$ times/week/at least 20 minutes), presence of smoking related diseases such as cardiovascular diseases, cancer, or chronic obstructive pulmonary disease (COPD), passive smoking at home, psychiatric comorbidity, other drugs dependence, and pulmonary function, according to the official statement of the American Thoracic and European Respiratory Societies [19]. Adjusted values by GLI equations were obtained for the following spirometry parameters: forced vital capacity (FVC), forced expiratory volume in one second (FEV$_1$), and peak expiratory flow (PEF) were collected, and chronic airflow limitation, defined by an FEV1/FVC ratio $<0.70$ was calculated.

## Asthma related variables

The following data were collected in asthmatic patients: asthma symptoms (coughing, wheezing, chest tightness, and dyspnea), Asthma Control Test (ACT) questionnaire [20], exacerbation rate during the last month, treatments taken during the previous year including inhaled, systemic or oral steroids, short-(SABA) or long-(LABA) acting beta2 agonists, leukotriene modifiers, anticholinergics, biologics, and others.

## Smoking related variables

In all study participants we assessed: smoking regularity (regular *vs.* occasional), smoking characteristics (age of first cigarette and smoking onset, number of cigarettes per day and pack-years), smoking cessation attempts (number, relapse, causes of relapse), smoking cessation interventions (pharmacological and cognitive-behavior interventions), smoking cessation support, motivation to quit (0–10 scale), smoking self-efficacy to quit (0–10 scale), reward (positive or negative), Fagerström Tolerance Nicotine Dependence Questionnaire (FTND) [21], and UISPM test (Test de la Unidad del Instituto de Salud Pública de Madrid), carbon

monoxide (CO) in exhaled air and carboxyhemoglobin (HbCO) levels, use of health care resources (visits to the emergency room and hospital admissions).

UISMP is a test developed in Spain to evaluate six domains related with smoking: stimulation, sedation, automatism, social dependence, psychic dependence, and gestural dependence. This test has not been validated in English but in Spanish [22].

Pack-years was calculated by multiplying the number of cigarettes consumed per day by the number of years of tobacco use and the obtained figure was divided by 20.

The data were collected using a numerical code that did not allow the identification of the participant.

## Statistical analysis

First, a descriptive analysis of the study sample was performed. We used central tendency (mean and median) and dispersion (standard deviation and interquartile range) measures in the case of quantitative variables, depending on whether they were normally distributed. In the case of qualitative variables, frequency tables and percentage distribution were used.

Secondly, asthma frequency in the sample was estimated by calculating the proportion and its 95% confidence interval (CI), assuming a Poisson distribution.

Finally, both study groups (smokers with and without asthma) were compared using hypothesis contrast tests for continuous (Student's t or Mann-Whitney U) and categorical (chi-square) variables.

## Results

The final sample was composed of 329 smokers. Three of the participating units reached the sample size goal of 68 smokers. Nevertheless, recruitment was stopped because the sample size was sufficient to meet the study objectives.

Regarding sociodemographic characteristics, most smokers were caucasian (98%), 51% were women, with a mean age of 56±10 years old. Thirty-six percent of the participants had a very low educational level (no schooling or elementary school), and 23% were university graduates. We found that 55% of participants were active workers, 75% lived in urban areas, and 58% were physically active (Table 1).

Table 2 shows asthma related variables. Among smokers, 18.6% (95% CI [14.3, 24.0]) presented an asthma diagnosis. Asthmatic patients displayed a mean ACT score of 17±5, and 15% reported exacerbations during the last month. The most frequent symptoms were dyspnea (34%), coughing (31%), and wheezing (13%). LABAs were the most frequently prescribed inhaler medications during the previous year (45%), followed by anticholinergics (36%), inhaled steroids (31%), and SABAs (30%). Only three patients (1%) had received biologic therapy (Table 2).

Compared to non-asthmatics (Table 3), asthmatic smokers were mostly women (73.8% *vs*. 46.0%, p<0.001), significantly younger (50.4 *vs*. 57.7, p<0.0001), displayed a lower COPD frequency (6.6% *vs*. 31.7%, p<0.001), and a higher prevalence of anxious-depressive syndrome (54.1% *vs*. 26.3%, p<0.0001). Up to 13% of the smokers studied had chronic airflow limitation, defined by an FEV1/FVC ratio in spirometry <0.70. Passive exposure to tobacco smoke at home was also more common in asthmatic smokers, although the difference did not reach statistical significance (49.2% *vs*. 37.9%, p = 0.104). Similarly, no significant differences were observed regarding the pulmonary function tests, nor in the presence of chronic airflow limitation between asthmatic and no asthmatic smokers.

The comparison of smokers with and without asthma showed no significant differences in terms of habit type, age of first cigarette or of smoking onset, and number of cigarettes per day

**Table 1. Sociodemographic characteristics of the study sample (n = 329).**

| Variable | n (%) |
|---|---|
| Smoking Units | |
| • Clinical University Hospital, Santiago de Compostela | 68 (20.7%) |
| • Clinical University Hospital, Málaga | 63 (19.1%) |
| • Clinical University Hospital San Pedro de Alcántara, Cáceres | 68 (20.7%) |
| • Clinical University Hospital, Valencia | 62 (18.8%) |
| • Clinical University Hospital, San Carlos, Madrid | 68 (20.7%) |
| Caucasian ethnicity | 323 (98.5%) |
| Gender (female) | 169 (51.5%) |
| Educational level | |
| • No education/Primary school | 116 (35.6%) |
| • High School | 66 (20.2% |
| • Elementary professional training | 41 (12.6%) |
| • Superior professional training | 27 (6.3%) |
| • University | 76 (23.3%) |
| Employment status | |
| • Unemployed | 28 (8.6%) |
| • Active | 180 (55.4%) |
| • Retired | 95 (29.2%) |
| • Student | 1 (0.3%) |
| • Housewife | 21 (6.5%) |
| Residence | |
| • Urban | 245 (75.4%) |
| • Residential area | 80 (24.6%) |
| Physical activity | |
| • None | 65 (19.9%) |
| • Mild (<5 times/week/at least 30 minutes) | 73 (22.4%) |
| • Moderate (≥5 times/week/at least 30 minutes) | 114 (35.0%) |
| • Intense (≥3 times/week/at least 20 minutes) | 74 (22.7%) |
| Age[†] (years) (n = 329) | 56.3±10.5 |

Otherwise indicated, results are expressed as numbers and percentages (%)

† Mean ± standard deviation.

(Table 4). However, non-asthmatic individuals had been smoking for longer than asthmatics (38±10 *vs.* 31±12, p<0.0001) and had consumed a greater number of packs/year (45±25 *vs.* 32 ±20, p = 0.0001). Although 306 patients reported having received smoking cessation treatment, only 263 (80.4%) stated they had tried to go for more than 24 hours without smoking. Even though no differences in smoking cessation attempts were observed between groups, non-asthmatics used smoking cessation treatment more commonly than asthmatics (49% *vs.* 30%, p = 0.010), mainly varenicline (30% *vs.* 17%, p = 0.044), had a higher relapse frequency (85% *vs* 74%, p = 0.029), better support to quit smoking (77% *vs.* 60%, p = 0.008), and a higher smoking self-efficacy (6±2 *vs.* 5±2, p = 0.035) (Table 4).

No differences in tobacco dependence, measured by the total FTND score or questions measuring psycho-social dependence, were observed between asthmatics and non-asthmatics. However, there were significant differences in FTND categories (p = 0.046), with higher frequency of the "low dependence" category in asthmatics (19% *vs.* 8%). The comparison of the individual items did not highlight significant differences between the two groups. Finally, CO

**Table 2. Asthma related variables.**

| Variable | n (%) |
|---|---|
| Asthma diagnosis | 61 (18.6%) |
| Exacerbations during the last month | 9 (15.2%) |
| ACT score[†] | 16.7±5.2 |
| Respiratory symptoms | |
| • Coughing | 108 (30.8%) |
| • Wheezing | 44 (13.4%) |
| • Chest tightness | 26 (7.9%) |
| • Dyspnea | 111 (33.7%) |
| Treatments during the previous year | |
| • Inhaled steroids | 81 (30.8%) |
| • Maintenance oral corticosteroids | 2 (0.8%) |
| • Systemic steroids | 3 (1.1%) |
| • Rescue short-acting beta2 agonists (SABAs) | 79 (30.0%) |
| • Long acting beta2 agonists (LABAs) | 119 (45.1%) |
| • Leukotriene modifiers | 24 (9.1%) |
| • Anticholinergics | 94 (35.6%) |
| • Omalizumab | - |
| • Mepolizumab | 2 (0.8%) |
| • Benralizumab | 1 (0.4%) |
| • Reslizumab | - |
| • Dupilumab | - |
| • Others | 48 (18.3%) |

Otherwise indicated, results are expressed as numbers and percentages (%).

† Mean ± standard deviation.

**Abbreviation**: ACT = Asthma Control Test

and carboxyhemoglobin levels, as well as health resource use (emergency room visits and admissions) were similar in asthmatics and non-asthmatics patients (Table 4).

## Discussion

In this project we have found a high asthma frequency in Spanish attending smoking cessation units. We have also described both smoking and asthma, in order to characterize the currently known "asthma smoking phenotype".

The global frequency of asthma in smokers attending smoking cessation units in Spain is almost 20%. The prevalence of asthma varies widely around the world, ranging from 0.2% to 21.0% in adults, probably due to phenotypic variability, different study populations, and diagnostic criteria [23, 24]. Most of the previous studies were population-based with self-reported or questionnaire-based diagnosis, whereas we have used a highly selected population (smokers attending smoking cessation units) with a diagnosis based on a documented asthma diagnosis evidence. It is worth mentioning that the observed higher frequency of asthmatics among women smokers is consistent with the literature [18, 25–28]. However, despite these limitations and comparison difficulties, these results are critical as both smoking and asthma present pathophysiological mechanisms involved in the development of inflammatory airway diseases and might present synergic or additive effects [29, 30]. In fact, several observational studies suggested that cigarette smoking significantly increases asthma risk and prevalence, especially

**Table 3. Demographic and clinical characteristics in smokers with and without asthma.**

| Variable | Total (n = 327) | No asthmatic (n = 266) | Asthmatic (n = 61) | P value |
|---|---|---|---|---|
| Age (years) † | 56.3±10.5 | 57.7±9.9 | 50.4±10.9 | **<0.0001** |
| Gender (female) | 167 (51.2%) | 122 (46.0%) | 45 (73.8%) | **<0.0001** |
| Passive smokers | 130 (40.0%) | 100 (37.9%) | 30 (49.2%) | 0.104 |
| Frequency of passive exposure | n = 290 | n = 232 | n = 57 | 0.101 |
| • None | 106 (36.5%) | 83 (35.8%) | 23 (40.3%) | |
| • <1 hour | 72 (24.8%) | 64 (27.6%) | 8 (14.0%) | |
| • ≥1 hour | 112 (38.6%) | 85 (36.6%) | 26 (45.6%) | |
| Smoking related diseases | | | | |
| • Cardiovascular | 125 (38.5%) | 103 (38.9%) | 22 (36.7%) | 0.752 |
| • COPD | 88 (27.0%) | 84 (31.7%) | 4 (6.6%) | **<0.0001** |
| • Cancer | 48 (14.8%) | 44 (16.7%) | 4 (6.7%) | 0.068 |
| • Others | 192 (60.2%) | 150 (57.7%) | 42 (71.2%) | 0.056 |
| Psychiatric comorbidity | | | | |
| • Anxiety-depression | 103 (31.5%) | 70 (26.3%) | 33 (54.1%) | **<0.0001** |
| • Bipolar disorder | 6 (1.8%) | 4 (1.5%) | 2 (3.3%) | 0.311 |
| • Schizophrenia | 6 (1.8%) | 5 (1.9%) | 1 (1.6%) | 1.000 |
| • Others | 35 (10.7%) | 28 (10.5%) | 7 (11.5%) | 0.829 |
| Other drugs dependence | | | | |
| • Heroin | - | | | |
| • Cocaine | 8 (2.4%) | 5 (1.9%) | 3 (4.9%) | 0.172 |
| • Designer drugs | - | | | |
| • Others | 37 (11.2%) | 32 (12.0%) | 5 (8.2%) | 0.394 |
| Lung function† | | | | |
| • FVC (%) | 101.7±79.6 | 104.0±92.2 | 95.8±23.7 | 0.764 |
| • $FEV_1$ (%) | 82.3±22.8 | 81.2±22.9 | 85.0±22.7 | 0.564 |
| • PEF (l/m) | 75.3±109.5 | 75.1±115.4 | 75.7±93.5 | 0.355 |
| • FEV1/FVC | 0.85±0.15 | 0.84±0.16 | 0.87±0.14 | 0.137 |
| • Chronic airflow limitation (%): (FEV1/FVC<0.70) | 27 (13.0%) | 21 (13.9%) | 6 (10.7%) | 0.545 |

*Otherwise indicated, results are expressed as numbers and percentages (%). Significant results are written in bold.

† Mean ± standard deviation.

**Abbreviations**: COPD = Chronic obstructive pulmonary disease; FVC = Forced vital capacity; $FEV_1$ = Forced expiratory volume in one second; PEF = Peak expiratory flow.

in females, as women were found to be more vulnerable to the effect of tobacco smoking compared with men [25–28, 31]. Therefore, in daily practice smokers should regularly be evaluated in order to rule out asthma.

The smoking habit in smokers with asthma was characterized by a lower use of cessation treatments and a lower self-efficacy to quit smoking compared with no-asthmatic smokers. Our results are consistent with those of a U.S. study involving smokers using quit lines, which found a significantly lower 30-day quit rate in smokers with asthma compared to those without chronic conditions [32]. A greater cessation failure rate addition in smokers with asthma has been observed by other authors [33]. On the other hand, the presence of asthma in smokers was accompanied by a higher anxiety level and other psychopathological traits, which may make it more difficult to quit smoking despite having a good motivation for smoking cessation [34, 35].

Table 4. Characterization of smoking by asthma status.

| Variable | Total (n = 327) | Non-asthmatic (n = 266) | Asthmatic (n = 61) | P value |
|---|---|---|---|---|
| **Smoking regularity** | | | | 0.777 |
| • Regular smoker | 304 (93.2%) | 246 (92.8%) | 58 (95.1%) | |
| • Occasional smoker | 22 (6.7%) | 19 (7.2%) | 3 (4.9%) | |
| Smoking characteristics[†] | | | | |
| • Age of first cigarette | 15.9±3.5 | 15.8±3.2 | 16.6±4.7 | 0.533 |
| • Age of smoking onset | 18.0±4.1 | 18.0±3.9 | 18.3±5.0 | 0.860 |
| • Smoking duration (years) | 37.1±12.1 | 38.5±10.5 | 31.1±12.1 | **<0.0001** |
| • Number of cigarettes per day | 18.9±11.8 | 19.0±11.9 | 18.5±11.2 | 0.643 |
| • Number of pack-years | 42.5±24.8 | 44.8±25.2 | 32.4±20.1 | **0.0001** |
| **Attempts to quit smoking** | | | | |
| • Any attempts | 263 (80.4%) | 216 (81.2%) | 47 (77.0%) | 0.461 |
| • Number of quit attempts‡ | 1 (1–1) | 1 (1–1) | 1 (0–1) | 0.697 |
| • Smoking cessation interventions | 139 (45.4%) | 121 (49.0%) | 18 (30.5%) | **0.010** |
| ○ Nicotine replacement therapy | 38 (12.4%) | 33 (13.4%) | 5 (8.5%) | 0.306 |
| ○ Varenicline | 84 (27.4%) | 74 (30.0%) | 10 (16.9%) | **0.044** |
| ○ Bupropion | 15 (4.9%) | 14 (5.7%) | 1 (1.7%) | 0.318 |
| ○ Cognitive-behaviour intervention | 8 (2.6%) | 6 (2.4%) | 2 (3.4%) | 0.653 |
| ○ Other | 12 (3.9%) | 9 (3.6%) | 3 (5.1%) | 0.707 |
| **Last quit attempt** | | | | |
| • Days without smoking‡ | 60 (7–240) | 60 (7–240) | 60 (3–240) | 0.495 |
| • Relapse | 251(83.1%) | 206 (85.5%) | 45 (73.8%) | **0.029** |
| • Causes of smoking relapse | | | | 0.780 |
| ○ Abstinence syndrome | 103 (41.7%) | 82 (42.6%) | 17 (37.8%) | |
| ○ Weight gain | 4 (1.6%) | 3 (1.5%) | 1 (2.2%) | |
| ○ Life crisis | 33 (13.4%) | 26 (12.9%) | 7 (15.6%) | |
| ○ Depression | 16 (6.5%) | 12 (5.9%) | 4 (8.9%) | |
| ○ Social causes | 91 (36.8%) | 75 (37.1%) | 16 (35.6%) | |
| **Smoking cessation support** | 233 (73.5%) | 197 (76.6%) | 36 (60.0%) | **0.008** |
| Type of support | | | | 0.257 |
| • Family | 200 (86.6%) | 167 (85.6%) | 33 (91.7%) | |
| • Work | 15 (6.5%) | 15 (7.7%) | - | |
| • Friends | 16 (6.9%) | 13 (6.7%) | 3 (8.8%) | |
| • Motivation score to quit smoking | 8.3±2.2 | 8.3±2.2 | 8.3±2.0 | 0.646 |
| • Self-efficacy score to quit smoking | 5.5±2.4 | 5.7±2.4 | 5.0±2.4 | **0.035** |
| **Reward** | | | | 0.573 |
| • Negative (relieving withdrawal) | 181 (55.7%) | 149 (56.4%) | 32 (52.5%) | |
| • Positive (for pleasure) | 144 (44.3%) | 115 (43.6%) | 29 (47.5%) | |
| **Dependence** | | | | |
| • FTND total score | 6.4±2.1 | 6.5±2.0 | 6.1±2.4 | 0.310 |
| • FTND categories | | | | **0.046** |
| ○ Mild dependence | 32 (10.1%) | 21 (8.1%) | 11 (19.0%) | |
| ○ Moderate dependence | 105 (33.2%) | 87 (33.7%) | 18 (31.0%) | |
| ○ High dependence | 179 (56.6%) | 150 (58.1%) | 29 (50.0%) | |
| • UISPM test | 30.3±13.8 | 30.4±13.7 | 29.7±14.5 | 0.613 |
| ○ Stimulation | 4.4±3.5 | 4.3±3.5 | 4.8±3.7 | 0.291 |
| ○ Sedation | 5.4±2.4 | 5.4±2.4 | 5.5±2.3 | 0.858 |
| ○ Automatism | 4.3±3.5 | 4.5±3.6 | 3.7±3.5 | 0.079 |

*(Continued)*

**Table 4.** (*Continued*)

| Variable | Total (n = 327) | Non-asthmatic (n = 266) | Asthmatic (n = 61) | P value |
|---|---|---|---|---|
| ○ Social dependence | 7.2±4.9 | 7.2±4.8 | 7.0±4.9 | 0.660 |
| ○ Psychic dependence | 6.2±3.6 | 6.2±3.6 | 6.1±3.9 | 0.827 |
| ○ Gestural dependence | 2.8±3.1 | 2.8±3.0 | 2.6±3.5 | 0.314 |
| CO levels exhaled air (ppm) | 16.8±11.0 | 16.8±11.3 | 17.2±10.3 | 0.703 |
| HbCo levels (%) | 2.5±2.3 | 2.4±1.3 | 2.9±4.1 | 0.663 |
| Emergency room visits‡ | 0 (0–1) | 0 (0–1) | 0 (0–1) | 0.298 |
| Hospital admissions‡ | 0 (0–1) | 0 (0–1) | 0 (0–0) | 0.135 |

*Otherwise indicated, results are expressed as numbers and percentages (%). Significant results are written in bold.

† Mean ± standard deviation; ‡ Median (25th-75th percentiles).

Abbreviations: FTND = Fagerström Test for Nicotine Dependence; UISPM = Test de la Unidad del Instituto de Salud Pública de Madrid; CO = carbon monoxide; HbCO = carboxyhemoglobin.

Our data show a high level of passive exposure at home (40%), although without significant differences between asthmatic and non-asthmatic smokers. The relationship between passive exposure and asthma is well known [27, 36, 37] and is important to consider since passive exposure contributes to the adverse health outcomes presented by smokers with asthma [13, 15, 24, 38]. Therefore, is important to involve family members to prevent smoking at home.

We would like to highlight the significantly lower smoking cessation support in asthmatic versus no asthmatic smokers (60% vs 77%) found in our study. The fact that smokers with asthma are not more likely to receive smoking cessation counseling than general smokers has already been observed in the literature [12]. These data could be explained, at least in part, by the fact that these patients smoked less, had been smoking for a shorter time, and had a lower level of dependence. On the other hand, our results are in line with those reported in the literature [39, 40], and no differences were found in the quit rates between both groups. Considering the impact of smoking on this lung disease, perhaps we should consider using other types of intervention, even more intensive, in the group of smokers with asthma. Thus, more efforts are needed to implement efficient strategies to achieve smoking cessation in the whole smoking population, but particularly in those with concomitant asthma wishing to stop smoking, regardless of the dependence level or other smoking features [41, 42].

Finally, the main characteristics of asthma in smokers were related to poor asthma control, as measured by the ACT score, as well as the high exacerbation frequency during the last month, the need for rescue medication, and frequent use of LABAs, inhaled steroids, and SABAs. These results are consistent with the literature. Indeed, evidence suggests that smoking is associated with decreased asthma control, increased exacerbation risk, worsened prognosis, and decreased inhaled corticosteroids effectiveness [6, 10, 12, 15, 18]. Asthmatic smokers constitute a subpopulation with higher risk of poor evolution requiring special attention and control.

Our work presents some limitations. The most important one is probably the representativeness of the sample. The smokers studied attended smoking cessation units and may therefore not be representative of the smoker population. This selection bias may be responsible for an overestimation of the outcomes studied. However, studying this type of selected smokers can be a great opportunity for the implementation of smoking cessation strategies given the reversible effect of pulmonary involvement.

In conclusion, asthma frequency is very high among smokers. Asthma screening in smokers and integral management of smokers with asthma are fundamental to prevent poor outcomes in this subgroup of smokers and to achieve smoking cessation.

## Supporting information

**S1 Checklist.**
(DOCX)

## Acknowledgments

The authors thank to InMusc for their methodology and statistical support.

## Author Contributions

**Conceptualization:** Juan-Antonio Riesco, Carlos Rábade, Jaime Signes-Costa, Eva Cabrera, Carlos-A Jimenez.

**Data curation:** Juan-Antonio Riesco, Carlos Rábade, Jaime Signes-Costa, Eva Cabrera, Carlos-A Jimenez.

**Supervision:** Juan-Antonio Riesco, Carlos Rábade, Jaime Signes-Costa, Eva Cabrera, Carlos-A Jimenez.

**Writing – original draft:** Juan-Antonio Riesco, Carlos-A Jimenez.

**Writing – review & editing:** Juan-Antonio Riesco, Carlos-A Jimenez.

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
