## [Decision Letter · Decision Letter 0]

18 Aug 2023

PONE-D-23-14534Frequency and Differential Characteristics of Asthma in SmokersPLOS ONE

Dear Dr. JA Riesco,

Thank you for submitting your manuscript to PLOS ONE. After careful consideration, we feel that it has merit but does not fully meet PLOS ONE’s publication criteria as it currently stands. Therefore, we invite you to submit a revised version of the manuscript that addresses the points raised during the review process. This is an original manuscript of interest for respiratory physicians, assessing the frequency and smoking characteristics of asthmatic patients attending 5 different smoking cessation units in Spain.

An adequate number of study subjects is analysed. However the reasearch needs to be improved, particularly in the analysis of lung function and outcome of smoking cessation.

We look forward to receiving your revised manuscript.

Kind regards,

Manlio Milanese

Academic Editor

PLOS ONE

Journal Requirements:

This study was supported by a grant of Menarini.

Additional Editor Comments:

The manuscript is original and of interest for respiratory physicians. However the reasearch needs to be improved, particularly in the analysis of lung function and outcome of smoking cessation. Please follow our suggestions.

Reviewers' comments:

Reviewer's Responses to Questions

**Comments to the Author**

1. Is the manuscript technically sound, and do the data support the conclusions?

Reviewer #1: Partly

2. Has the statistical analysis been performed appropriately and rigorously? 

Reviewer #1: Yes

3. Have the authors made all data underlying the findings in their manuscript fully available?

Reviewer #1: Yes

4. Is the manuscript presented in an intelligible fashion and written in standard English?

Reviewer #1: Yes

5. Review Comments to the Author

Reviewer #1: General comment

This is an original paper which assess the frequency and smoking characteristics of asthmatic patients attending 5 different smoking cessation units in Spain.

An adequate number of study subjects is analysed. The topic is of interest for respiratory physicians. The reasearch need to be improved, particularly in the analysis of lung function and outcome of smoking cessation.

Specific comments

Major

- The title should contain information on the fact that the study subjects are smokers attending smoking cessation units.

- Considering that the study subjects are smokers attending smoking cessation units, the study should include a multivariate analysis assessing the variables possibly associated with smoking cessation at least at short-medium term (e.g. 4-12 weeks from the start of the smoking cessation program) for asthmatic/non-asthmatic smokers.

- It is unclear if the presented data about the "Smoking cessation interventions" and "Relapse" (Table 4) are related to previous quit attempts or to the actual attempt at the moment of the enrollment in the study.

- The frequency of airway obstruction in the study subjects should be reported. Obstruction should be defined as FEV1/FVC < 5th percentile of the predicted value, by applying GLI equations (Stanojevic S et al. Eur Respir J 2022)

- Discussion section: "We would like to highlight the significantly lower smoking cessation support in asthmatic smokers found in our study." (page 15, lines 263-264). This affirmation should be further clarified in relation to the presented results.

- Discussion section: "However, these patients had significant pulmonary involvement regardless of smoking and did not achieve good smoking cessation rates." (page 15, lines 266-267). This affirmation should be further clarified in relation to the presented results and to those to be added in a revised version of the manuscript.

Minor

- The abstract lacks of a clear aim of the study.

- "packs per year" should be replaced by: pack-years, throughout the manuscript. In the Material and methods section it should be reported how "packs per year"/pack-years were calcutated.

- Material and methods section. The used lung function protocol should be reported.

- Material and methods section. Reference and description of the UISPM=Test de la Unidad del Instituto de Salud Pública de Madrid should be reported.

- In the title of Table 1, 2, 3 and 4 it should be reported the total number of the study subjects (n=329). Within each table, a note reporting that there were missing data for some indicated variable should be added.

- "Table 2" is not quoted in the text of the Results section.

- Table 2. It should be clarified that Table 2 refers only to the smokers with diagnosis of asthma.

- Title of Table 3: correct "Tabla 3".

- Table 3. Those reported in this table are clinical and respiratory function characteristics in addition to demographic characteristics. The title should be corrected accordingly.

- Table 3. FEV1, FVC, and PEF lack of units of measurement.

- Table 3: numbers and percentages of the obstructed study subjects should be added.

- Table 3. Information on cannabis use are lacking.

- Table 4. "CO levels (exhaled air)" and "HbCo levels" lack of units of measurement.

6. PLOS authors have the option to publish the peer review history of their article (what does this mean?). If published, this will include your full peer review and any attached files.

Reviewer #1: **Yes: **Francesco Pistelli

---

## [Author Response · Author response to Decision Letter 0]

3 Oct 2023

PONE-D-23-14534

Frequency and Differential Characteristics of Asthma in Smokers

Dear Editor and reviewers,

We would like to thank you for all the constructive comments and suggestions. We really appreciated them and are confident that will clearly contribute to improving the manuscript. 

See below our responses (in blue) to your comments (in black) and applied changes.

Looking forward to your response,

Juan Antonio Riesco on behalf of the authors

REVIEWER #1: 

General comment

This is an original paper which assess the frequency and smoking characteristics of asthmatic patients attending 5 different smoking cessation units in Spain.

An adequate number of study subjects is analysed. The topic is of interest for respiratory physicians. The research needs to be improved, particularly in the analysis of lung function and outcome of smoking cessation.

Specific comments. Major

- The title should contain information on the fact that the study subjects are smokers attending smoking cessation units.

We agree with the reviewer; we have modified the title according to his suggestion. The current title is:

Title page: Frequency and characteristics of Asthma in smokers attending smoking cessation units in Spain. 

- Considering that the study subjects are smokers attending smoking cessation units, the study should include a multivariate analysis assessing the variables possibly associated with smoking cessation at least at short-medium term (e.g. 4-12 weeks from the start of the smoking cessation program) for asthmatic/non-asthmatic smokers.

Thanks for the comment. The analysis of the factors associated with smoking cessation and possible differences in asthmatics vs non-asthmatics can be very interesting. However, these factors are not the focus of the study and, therefore, information on possible determinants or confounding factors of smoking cessation (such as duration of treatments, adherence, etc.) may be incomplete.

- It is unclear if the presented data about the "Smoking cessation interventions" and "Relapse" (Table 4) are related to previous quit attempts or to the actual attempt at the moment of the enrollment in the study.

We agree that Table 4, on differential characteristics of smoking between asthmatics and non-asthmatics, is difficult to understand. To make it easier to read, we have modified the table by grouping related variables under the same characteristic (in bold type). Thus, the variables related to any quit attempt are grouped together, as well as those referring to the last attempt, regularity of the habit, reward, dependence, or others. 

- The frequency of airway obstruction in the study subjects should be reported. Obstruction should be defined as FEV1/FVC < 5th percentile of the predicted value, by applying GLI equations (Stanojevic S et al. Eur Respir J 2022)

The spirometer software has the GLI equations implemented, so the results are already expressed as adjusted. However, we agree on the importance of adding some additional information. In this sense, we have calculated the FEV1/FVC ratio and the percentage of patients with chronic airflow limitation, defined by an FEV1/FVC ratio <0.70. (Page 11). 

- Discussion section: "We would like to highlight the significantly lower smoking cessation support in asthmatic smokers found in our study." (page 15, lines 263-264). This affirmation should be further clarified in relation to the presented results.

Thanks for the comment. This sentence is based in results shown in table 4, with a lower proportion of support to quit smoking in asthmatics versus no-asthmatics (60% vs. 77%). On the other hand, we have added some bibliographic reference in the discussion to support this result (page 16).

- Discussion section: "However, these patients had significant pulmonary involvement regardless of smoking and did not achieve good smoking cessation rates." (page 15, lines 266-267). This affirmation should be further clarified in relation to the presented results and to those to be added in a revised version of the manuscript. 

The reviewer is right. The sentence is confusing. The idea is to emphasize that the absence of differences in quit rates between asthmatics and no asthmatics probably points to the need to use more intensive intervention strategies in asthmatic smokers because of the impact of smoking on the asthmatic patient's lung function. 

We have added a short explanation in this regard, as well as bibliography that supports our results (page 16).

Minor

- The abstract lacks a clear aim of the study.

The reviewer is right, thank you very much for the comment. We have added a sentence with the main purposes of the study (page 2). 

- "packs per year" should be replaced by: pack-years, throughout the manuscript. In the Material and methods section, it should be reported how "packs per year"/pack-years were calcutated.

Thank you very much for the comment. We have added a sentence on how to calculate the number of pack-years in the methods section (page 8; line 154). In addition, we have changed the expression “packs per year” by “pack-years” throughout the text, as suggested by the reviewer. 

- Material and methods section. The used lung function protocol should be reported.

The reviewer is right. Thank you for your comment. We have added the requested information and referenced it in the relevant section (page 7). 

- Material and methods section. Reference and description of the UISPM=Test de la Unidad del Instituto de Salud Pública de Madrid should be reported.

Indeed, we agree that the information on this test is scarce in the article. Consequently, we have added a short explanation of this tool as well as the corresponding bibliography in Methods section (page 8).

- In the title of Table 1, 2, 3 and 4 it should be reported the total number of the study subjects (n=329). Within each table, a note reporting that there were missing data for some indicated variable should be added.

Certainly, the n changes in different variables due to missing values (that is why the n of each variable was placed in each row). In order to lighten the table, the partial n's have been suppressed and a sentence about the existence of missing values has been added.

- "Table 2" is not quoted in the text of the Results section.

Explanation of table 2 has been quoted in the text (page 10).

- Table 2. It should be clarified that Table 2 refers only to the smokers with diagnosis of asthma.

Thanks for the comment. An explanatory sentence has been added before describing table 2 (page 10).

- Table 3: 

• Title correct "Tabla 3". The spelling error has been corrected (page 12) 

• Those reported in this table are clinical and respiratory function characteristics in addition to demographic characteristics. The title should be corrected accordingly. Title has been modified according to the reviewer’s suggestion (page 11).

• FEV1, FVC, and PEF lack of units of measurement. Corresponding units have been added to respective parameters in table 3. 

Numbers and percentages of the obstructed study subjects should be added. 

The chronic air flow limitation, defined by a FEV1/FVC ratio <0.70, has been calculated and added to lung function parameters in table 3. 

Information on cannabis use is lacking.

No specific information on cannabis use was collected.

- Table 4. "CO levels (exhaled air)" and "HbCo levels" lack of units of measurement. 

The corresponding units have been added to the table (ppm for CO levels and % for HbCO levels) (page 14).

---

## [Decision Letter · Decision Letter 1]

6 Nov 2023

Frequency and characteristics of Asthma in smokers attending smoking cessation units in Spain

PONE-D-23-14534R1

Dear Dr. Riesco,

We’re pleased to inform you that your manuscript has been judged scientifically suitable for publication and will be formally accepted for publication once it meets all outstanding technical requirements.

Kind regards,

Manlio Milanese

Academic Editor

PLOS ONE

Additional Editor Comments (optional):

Reviewers' comments:

Reviewer's Responses to Questions

**Comments to the Author**

1. If the authors have adequately addressed your comments raised in a previous round of review and you feel that this manuscript is now acceptable for publication, you may indicate that here to bypass the “Comments to the Author” section, enter your conflict of interest statement in the “Confidential to Editor” section, and submit your "Accept" recommendation.

Reviewer #1: All comments have been addressed

2. Is the manuscript technically sound, and do the data support the conclusions?

Reviewer #1: Yes

3. Has the statistical analysis been performed appropriately and rigorously? 

Reviewer #1: Yes

4. Have the authors made all data underlying the findings in their manuscript fully available?

Reviewer #1: Yes

5. Is the manuscript presented in an intelligible fashion and written in standard English?

Reviewer #1: Yes

6. Review Comments to the Author

Reviewer #1: The authors have adequately addressed the comments raised by this reviewer and the manuscript is now improved.

7. PLOS authors have the option to publish the peer review history of their article (what does this mean?). If published, this will include your full peer review and any attached files.

Reviewer #1: No

---

## [Editor Report · Acceptance letter]

27 Nov 2023

PONE-D-23-14534R1 

Frequency and characteristics of Asthma in smokers attending smoking cessation units in Spain 

Dear Dr. Riesco:

I'm pleased to inform you that your manuscript has been deemed suitable for publication in PLOS ONE. Congratulations! Your manuscript is now with our production department. 

Kind regards, 

on behalf of

Dr. Manlio Milanese 

Academic Editor

PLOS ONE